# Challenges and Opportunities in Clinical Applications of Blood-Based Proteomics in Cancer

**DOI:** 10.3390/cancers12092428

**Published:** 2020-08-27

**Authors:** Ruchika Bhawal, Ann L. Oberg, Sheng Zhang, Manish Kohli

**Affiliations:** 1Proteomics and Metabolomics Facility, Institute of Biotechnology, Cornell University, Ithaca, NY 14853, USA; rb822@cornell.edu; 2Division of Biomedical Statistics and Informatics, Mayo Clinic, Rochester, MN 55905, USA; Oberg.Ann@mayo.edu; 3Department of Medicine, Division of Oncology, Huntsman Cancer Institute, University of Utah, Salt Lake City, UT 84112, USA

**Keywords:** serum, proteomics, mass spectrometry, biomarkers

## Abstract

**Simple Summary:**

The traditional approach in identifying cancer related protein biomarkers has focused on evaluation of a single peptide/protein in tissue or circulation. At best, this approach has had limited success for clinical applications, since multiple pathological tumor pathways may be involved during initiation or progression of cancer which diminishes the significance of a single candidate protein/peptide. Emerging sensitive proteomic based technologies like liquid chromatography mass spectrometry (LC-MS)-based quantitative proteomics can provide a platform for evaluating serial serum or plasma samples to interrogate secreted products of tumor–host interactions, thereby revealing a more “complete” repertoire of biological variables encompassing heterogeneous tumor biology. However, several challenges need to be met for successful application of serum/plasma based proteomics. These include uniform pre-analyte processing of specimens, sensitive and specific proteomic analytical platforms and adequate attention to study design during discovery phase followed by validation of discovery-level signatures for prognostic, predictive, and diagnostic cancer biomarker applications.

**Abstract:**

Blood is a readily accessible biofluid containing a plethora of important proteins, nucleic acids, and metabolites that can be used as clinical diagnostic tools in diseases, including cancer. Like the on-going efforts for cancer biomarker discovery using the liquid biopsy detection of circulating cell-free and cell-based tumor nucleic acids, the circulatory proteome has been underexplored for clinical cancer biomarker applications. A comprehensive proteome analysis of human serum/plasma with high-quality data and compelling interpretation can potentially provide opportunities for understanding disease mechanisms, although several challenges will have to be met. Serum/plasma proteome biomarkers are present in very low abundance, and there is high complexity involved due to the heterogeneity of cancers, for which there is a compelling need to develop sensitive and specific proteomic technologies and analytical platforms. To date, liquid chromatography mass spectrometry (LC-MS)-based quantitative proteomics has been a dominant analytical workflow to discover new potential cancer biomarkers in serum/plasma. This review will summarize the opportunities of serum proteomics for clinical applications; the challenges in the discovery of novel biomarkers in serum/plasma; and current proteomic strategies in cancer research for the application of serum/plasma proteomics for clinical prognostic, predictive, and diagnostic applications, as well as for monitoring minimal residual disease after treatments. We will highlight some of the recent advances in MS-based proteomics technologies with appropriate sample collection, processing uniformity, study design, and data analysis, focusing on how these integrated workflows can identify novel potential cancer biomarkers for clinical applications.

## 1. Introduction

An accumulation of genetic and epigenetic alterations that change protein expression can lead to tumorigenesis and the aggressiveness of cancer post diagnosis. This complexity of tumorigenesis and cancer progression has been rationalized in several hallmarks of cancer [1]. In addition, clonal evolutionary processes after tumorigenesis appear to occur on an average of 1–10 mutations per cell division [2]. Some of these mutations result in functional and structural alterations in protein synthesis, which in turn may influence the natural history of cancer progression or the responsiveness of a tumor to treatments and interventions. For example, somatic mutations in leukemia-associated driver genes result in the expansion of a genetically identical clone of marrow and blood cells that results in the development of overt neoplasia at the rate of 0.5–1 percent of all cases. This clonal hematopoiesis of indeterminate potential (CHIP) [3] illustrates that some but not all alterations are functional or impact disease outcomes. Recently, the Pan Cancer Analysis of the Whole Genome (PCAWG) Consortium of the International Cancer Genome Consortium (ICGC) and The Cancer Genome Atlas (TCGA) reconstructed the life history and evolution of driver mutational sequences in 2778 cancers from 38 tumor types [4]. The phylogenetic evolutionary tree of most cancers appears to be characterized by early mutations in a constrained set of driver genes and is then followed by the continuous diversification of the mutational spectrum, leading to increased genomic instability in later cancer stages. Protein expression and functions are dependent on the transcript levels of driver genes, translational efficiency, regulated degradation, post-translational modifications (PTMs), and protein–protein interactions. The identification of changes in protein-signaling networks can help us understand underlying dynamic biological processes that lead to cancer progression and consequently identify biomarkers for disease management [5]. The characterization of these biological processes and complexities at the protein level is clearly needed. Serum/plasma proteomics has great potential in this regard, but has not been well elucidated for clinical applicability in developing multi-analyte algorithms to capture and incorporate biologically relevant clonal evolution based on serial profiling.

Since proteins are not easily synthesized or replicated like DNA in vitro, and exist in a wide range of concentrations, it is analytically challenging to characterize them. Highly sensitive mass spectrometric (MS)-based proteomic technologies have been developed to identify protein-based markers for cancer diagnosis, minimal residual disease monitoring, drug response prediction, prognostication, and the identification of novel therapeutic targets [6,7,8,9]. Broadly, the two stages of proteomics strategies that are currently used in cancer research are global quantitative proteomics and targeted quantitative proteomics. Global comparative proteomics deals with cataloguing and quantifying the abundance of proteins, protein modifications, or protein complexes in a given tissue or body fluid at a given time. Here, the patterns of protein expression are quantitatively measured and compared between cancer and non-cancer samples. This enables the researcher to identify potential candidate biomarkers for clinical applications [10,11,12]. Most studies based on global comparative proteomics have discovered hundreds of proteins present at differential abundances in sample groups depending upon the sample type. However, the validation of findings has been lacking, in part owing to the variance in large-scale proteome screening data caused by tumor heterogeneity and flaws in the design of the study. Targeted quantitative proteomics, on the other hand, focuses on the known candidate protein/disease markers and quantitatively measures their differential abundance between the cancer and non-cancer states in a relatively large set of samples [13,14]. Targeted quantitative proteomics mostly relies on the single/multiple reaction monitoring (SRM/MRM) method, which is faster, multiplexing large numbers of targets, and has a high sensitivity and a good degree of correlation with traditional antibody-based targeted analysis [15]. However, the sensitivity and dynamic range of SRM even under the most advanced MS cannot span or characterize the entire proteome of any given organism, and the development of a mass spectrometry assay for each of the target biomarkers is not cost effective.

A big analytical challenge in proteomic research for the characterization and validation of the proteome biomarker candidates is to deal with the dynamic nature and complexity of the cellular proteomes from different biological specimen types, such as blood, serum, plasma, and tissue. Novel MS-based analytical platforms and technologies are being constantly developed [16], which can interrogate tissue as well as fluid-based specimens. Human serum/plasma has a vast array of proteins and complexes, which makes it an attractive diagnostic platform for disease-associated biomarkers with clinical applications. It has advantages such as its minimal cost and ease of serial sample collection and processing. The growing interest in serum/plasma proteomics has led to a large number of biomarkers identified by MS-based discovery for different diseases, including cancer [17,18,19,20]. However, serum/plasma proteomics also brings unique challenges for biomarker discovery, which include experimental design, critical strategies for sample preparation and data acquisition, and data mining tools [21,22]. This paper will review advanced MS-based proteomics technologies and the clinical impact of proteomics biomarkers, and will focus on the recent development of proteomic strategies for serum/plasma specimens. Finally, we discuss future perspectives for MS-based proteomics and its clinical impact on the management of disease.

## 2. General Considerations for Serum/Plasma Proteomics-Based Biomarker Discovery

### 2.1. Choice of Blood Plasma or Serum for Proteomics Studies

Blood is the most commonly used medium for clinical laboratory assays and is an important circulating biofluid for the proteomic discovery of putative biomarkers used in the diagnosis and prognosis of many diseases. Blood samples are reasonably easy to obtain and process, are mostly considered as homogeneous, and have less variation among the healthy samples compared to other body fluids [23]. Serum/plasma are the most commonly used blood fractions for cancer biomarker development [24]. One of the most commonly asked questions for proteomic analysis is whether to use serum or plasma from blood. Plasma samples are obtained by the addition of an anticoagulant such as EDTA and citrate for proteomic analysis, and subsequent centrifugation to remove the cellular material from the collected blood. It is a challenging biological matrix, due to the large dynamic range of protein abundance predominated with albumin, immunoglobulin, and fibrinogens. Based on the Plasma Proteomic Project by the Human Proteome Organization (HUPO), the recommendation is the use of plasma over serum because of the integrity of the “raw” plasma with the expected lower degree of ex vivo degradation [25]. Serum samples, on the other hand, are obtained by withdrawing blood in the absence of any anticoagulants, allowing for the formation of a fibrin clot. Then, serum samples are centrifuged to remove blood cells, the fibrin clot containing a large portion of the fibrinogen content, and the platelets, which results in the qualitative difference between plasma and serum samples. Studies have reported that other proteins are also removed by specific or nonspecific interactions within the fibrin clot [26]. In addition, it was found that, due to removal of fibrinogen, certain proteins are inversely increased in serum, such as the levels of platelet-secreted vascular endothelial growth factor (VEGF) [27]. Despite the fact that the protein concentration of serum is only 3–4% less than that of plasma, prolonged clotting times may affect the serum proteome composition [28]. Hence, the choice of a sample type and preparation method has to be very specific and determined based on biomarker discovery in specific projects, but also the sample-collection protocol (e.g., the type of collection tube) and the sample-processing procedure (e.g., coagulation temperature, time allowed for coagulation, and anticoagulant used) are very important for removing any bias [29].

### 2.2. Important Factors for Designing Proteomic Studies for Cancer Biomarker Discovery

To begin with initial serum/plasma marker discovery, the clinical application question has to be precisely defined [30] in the context of the application of the test and the population of interest (Table 1). A marker can be for disease screening or early detection, where the focus is on detecting disease in a high-risk or highly susceptible patient population while still asymptomatic. Alternatively, it may be a diagnostic cancer marker, which provides an indication of the presence of disease in a symptomatic patient. A disease-monitoring marker indicates disease burden during treatment or monitoring. A prognostic cancer marker gives an indication of future patient outcome regardless of treatment. A treatment selection or predictive biomarker gives an indication of the expected outcome of a specific treatment. Thus, each is intended for use in a different patient population, requires specimens collected at different time points in the course of disease, and thus requires different study designs for marker discovery and validation. The disease incidence and prevalence, as well as the cost of false positive or false negative results, will drive the needed marker performance metrics of sensitivity and specificity. For example, general population screening for pancreatic cancer, a very deadly but low-prevalence disease, would require a specificity and sensitivity of at least 99% and 60%, respectively, to simultaneously be worthwhile and minimize the undue stress of false positive diseases [31]. On the other hand, specificity requirements can be lower for a disease with a higher prevalence. Guidance for determining ideal performance metrics in light of disease prevalence and the cost/benefit ratio is available, and markers with varying performance characteristics are described herein in later cancer-specific sections [32]. For developing proteomic-based predictive/prognostic/monitoring biomarkers, the challenges go beyond the prevalence of disease and will also need to take into account the type of targeted therapy; response rates to treatment; and time to event analysis, such as progression-based survival and death. The context of the downstream clinical application therefore will need to be included during the discovery design phase of development, but confirmed using independent validation cohorts.

The relevance and population of interest is critical to guide choice of study design and specimens for use in discovery [33]. Marker discovery work is best performed using samples collected at the stage in which the marker is intended for use (Table 1) [34,35,36,37]. For example, new-onset diabetes can be a prodromal illness for pancreatic cancer in some people up to three years prior to cancer diagnosis. In patients with newly diagnosed maturity onset diabetes, serum proteomics could be used to identify a detection marker of pancreatic cancer with the intention to screen patients newly diagnosed with diabetes for pancreatic cancer [38,39]. Marker discovery would ideally be performed using specimens collected at the diagnosis of diabetes in two groups of people—those who go on to have pancreatic cancer in the next three years (cases) and those who do not (controls). Differences in the protein abundance of any potential cancer markers between these two groups of patients would admittedly be expected to be small, but will truly represent the state of disease. Unfortunately, using convenience samples of specimens for such studies, while potentially generating initial exciting results, has typically led to unreliable markers of disease. This is likely due to a variety of pre-analytical sample handling methods and both known and unknown differences in case and control patient populations causing biases [40,41,42,43]. The same is true for attempting to discover markers of early stage disease in the extremes of the disease spectrum, such as healthy and end-stage disease [37]. Case and control specimens should be selected randomly from the same biobank of patients recruited prior to developing disease to ensure similar recruitment practices and specimen handling, and avoid biases in patient recruitment [44]. Such a retrospective nested case control study performed in prospectively collected specimens is known as the PRoBE (prospective-specimen collection, retrospective-blinded-evaluation) design in the biomarker literature. It highlights the need for establishing large biobanks of specimens collected prospectively and using uniform processing protocols prior to knowing eventual case or control status to enable such studies. This recognition has led to establishing such biobanks to support proteomic platforms for discovery and validation [41,44].

## 3. Pre-Analytic Challenges and Opportunities in Serum/Plasma Proteomics Cancer Biomarker Development

The serum/plasma generated from whole blood contains a wide range of proteins and small molecules secreted from various types of cells and tissues. There is an extraordinarily wide dynamic range of the serum/plasma proteins in terms of their abundance. In the serum/plasma from cancer patients, the abundance of clinically useful protein biomarker associated with cancer is typically at least 1000-fold lower than that of the commonly expressed highly abundant proteins. This leads to a huge analytical hurdle to identify clinically useful markers in serum/plasma samples. Other factors also can influence the discovery of serum-based markers in cancer patients, such as the pre-analytical processing of serum and the storage conditions of biospecimens. For example, serum samples stored at −20 °C versus −80 °C and processed at 4 °C versus 22 °C can impact the protein stability and subsequent marker discovery [45]. Serum specimens that undergo freeze-thaw cycles, which lead to the degradation of some proteins, may have poor outcomes for serum proteomic profiling. Enzymatic degradation can further complicate serum/plasma proteomics, as there may be a further loss of biomarkers in the serum/plasma, although this can be mitigated by adding protease inhibitors earlier during sample processing. It is critical to recognize and address these pre-analytical challenges accordingly. Some of these challenges can be overcome by techniques that employ the immunological depletion of high-abundance proteins such as albumin and immunoglobulins; the deployment of extensive fractionation using orthogonal three-dimensional protein separation to reduce complexity; and the targeted enrichment of specific subgroups of proteins or peptides of interest, such as glycoproteins and cysteine-rich proteins [46].

Despite these challenges, the application of the latest MS-based proteomic technologies to uniformly collected, prepared, and appropriately stored serum/plasma specimens offers great potential for cancer biomarker discovery. Uniformly applied pre-analytical strategies in serum proteomics has shown the successful identification of low-abundance proteins in blood, such as EGFR (1.3–3.5 µg/mL) and the hepatocyte growth factor activator (400 ng/mL) [47]. In ovarian cancer, diagnostic, prognostic, and predictive markers have similarly been reported using the comparative proteomic analysis of serum from ovarian cancer patients and healthy women. Sun et al. have found there are differentially expressed proteins with a higher sensitivity and specificity for prostate cancer patients [19,48,49,50]. This has led to intense MS-based research in other tumor types to identify proteins at very low concentrations for the potential discovery of cancer biomarkers.

## 4. Experimental Design and Statistical Considerations in Serum/Plasma Proteomics

It is important to study an adequate number of specimens. It is difficult to distinguish true biological differences from random and/or assay variability with small sample sizes [51]. Sample size planning involves the consideration of the expected differences between cases and controls, the desired power (chance of detecting true differences), the desired control over false discoveries (declaring significant differences that are actually null), technical variation, and biological variation. Chang et al. provide insights regarding sample size requirements for SRM measurements based on SRM variability in plasma from human ovarian cancer patients, and demonstrate that approximately 50 cases and 50 controls would be required to detect a fold change in a specific protein of 1.2 with 80% power and a 5% type I error rate [52]. If serum is the focus, these estimates can serve as a starting point for planning pilot studies in serum to evaluate variability for use in full study planning [53]. Fortunately, tools exist to assist in assessing these study design parameters during the study planning stages, and authors are starting to report information regarding technical and biological variability in addition to fold change, enabling educated parameter estimates for use in calculations [52,54,55,56,57].

While it is tempting to take short-cuts and proceed with small numbers or pool specimens due to the time- and resource-intensive nature of proteomics experiments, it has been shown that it is possible to proceed with larger sample sizes with the appropriate use of controls, replication, and statistical analysis tools for quantification and hypothesis testing [53,58]. The pooling of specimens prior to mass analysis without the specimens being labeled should be avoided, as it relies on the assumption of biological averaging and constrains the types of analyses that can be performed [59]. It is important to utilize randomization at all steps to protect from hidden or unknown biases, both when selecting samples to be studied and when allocating their specimens to the assay run order. Statistical experimental design can be especially useful here. The use of what is called a Randomized Block Design ensures that cases and controls are balanced in the assay run order [59,60,61]. For example, in a study comparing two groups using a label-free MS protocol, randomly allocating one specimen from each group in blocks of two in the run order prevents drift over time from confounding the group effects. Similarly, in a labeled protocol such as 8-plex isobaric tags for relative and absolute quantification (iTRAQ), randomly [59,62] assigning four specimens from each study group to each iTRAQ run as the authors describe accomplishes the same goal [22]. Other designs, such as a Split Plot or Latin Square, can be used to control for conditions that are difficult to randomize [63]. Labeled workflows have been shown to be more accurate in quantification, while label-free workflows are less expensive and have a higher throughput [52]. Note that even in labeled workflows, analysts should not generate ratios within runs prior to downstream analysis [52]. Rather, they should rely on the statistical model which incorporates the experimental design to perform the proper comparisons [52].

Statistical inference is crucial for distinguishing true differences in protein abundance from the natural biological variability and machine noise [57,59,64,65]. Choi et al. evaluated several statistical workflows in terms of the precision and accuracy of detecting true differential abundance [64]. They found that one of the most important determinations to guard against false discoveries was the use of methods that compare the biological signal to the amount of variation in the data. This is precisely what a *t*-test does: (measure of signal)/(measure of noise). There are statistical packages tailored for proteomic data which incorporate this at a more general level, with applicability to more than two groups, randomized block designs, time-course experiments, and more, appropriately accounting for the randomization process [54,66]. While intensity-based measurements of protein abundance are most common, spectral count methods are useful as well, and both can be used to successfully identify the differential abundance between groups, provided that some adjustment of multiple comparisons is performed [64]. The accuracy of differential expression findings was also found to be highly associated with the data analyst experience, highlighting the complexity of data analysis and the need for mentoring in the field [64]. The identity of interesting proteins or peptides discovered via global proteomic methods should be verified using targeted proteomic methods prior to further validation [24].

It is frequently desired to develop a panel of markers or signature that together performs better than a single marker alone, for which a complete study design and analysis plan should be finalized prior to beginning data collection [30]. Many computational methods are available for determining mathematical functions combining multiple proteins, peptides, or other ‘omics markers [67,68]. Performance is measured by metrics of accuracy (correct predictions), including receiver operating characteristic (ROC) area under the curve (AUC), sensitivity, specificity, and overall accuracy, with the specific metric driven by the study goals. The sample size requirements for this work are dependent primarily in the magnitude of differences, the number of proteins or markers used, and the proportion of cases to control [69,70,71]. Common regression modeling and machine learning tools used to develop such models generate the best fit to the data used, and thus generally yield overly optimistic estimates of model performance and are less likely to be validated. As a result, methods employing the penalization or shrinkage of regression parameter estimates or some other form of optimism correction should be employed during model development [72]. Lasso and elastic net are two such penalized regression tools that have been developed specifically for high-dimensional data with correlated features, as we see in proteomics [73,74]. The modeling results must be validated both internally with the data used to build the model, and externally with data generated from completely independent patients in order to ensure the reproducibility of the findings (Table 1) [41,75,76]. Internal validation can be performed via resampling methods such as cross validation or bootstrapping. These methods rely on repeatedly sampling a subset of study patients for use in the model development (training), and then applying the fitted model to the remaining patients (testing) to generate more realistic estimates of how the panel will perform in new datasets [77,78,79]. External validation is performed by applying this panel in a completely different set of patients, ideally from a different institution, and possibly even from a different country [30,72]. External validation is the strongest level of validation possible, and the panel algorithm and decision rules must be completely locked down before this level of validation is performed. The following references are excellent examples of use of internal and/or external validation in proteomic discovery work [80,81]. Guidelines are available for the full reporting of marker panel discovery and validation efforts [82,83,84,85].

## 5. Global Comparative Proteomics versus Targeted Quantitative Proteomics for Serum/Plasma Cancer Biomarker Discovery

Shotgun-based comparative proteomics technologies have emerged as the preferred tool, promising the systematic detection of discriminatory protein targets and their characteristic PTMs that collectively constitute molecular fingerprints secreted into blood or other tissues, reflecting the presence of cancer and the disease phenotype [62,86,87,88] (Figure 1). The differentially expressed proteins reflect their functions and the associated pathways that are involved, and can potentially be used for early diagnostic intervention. Global comparative proteomics with either labeled or label-free approaches have been widely used to compare the protein expression pattern across the samples, as well as to find out the therapeutic targets for the diagnosis (Figure 1) [89,90].

Label-free quantification is based on the signal intensity created by the peptides in the mass spectrometer. Those peptide ions with the same mass, charge, and retention time for each sample are compared and quantified on the basis of the difference in the peptide signal intensity that reflects the protein abundances in complex samples [91,92]. After a nanoLC-MS/MS analysis of tryptic digests from individual protein samples, generally more than 20,000 circulating peptides can be identified in human plasma. Two or more uniquely quantified peptides per protein are generally required to certify the confident identification and accurate quantitation of the target proteins. Recently, some label-free quantitation methods for in-depth serum proteomics have been used for the diagnosis of pancreatic cancer [93,94]. For label-free approaches, it is extremely important to generate reproducible patterns across all samples and variable time points and to develop software tools that can reliably align and match the related patterns across samples [95].

One of the most widely used chemical labeling techniques for global comparative proteomics is isobaric tags for relative and absolute quantification (iTRAQ) or tandem mass tag (TMT), which was used for labeling free amine groups at the peptide level [96]. TMT is an MS/MS-based strategy using isobaric labels for the accurate quantification of peptides and proteins from multiple samples within a single run [97]. The isobaric tags consist of an amine-specific reactive group, a balance group and a reporter group that provides a mass signature. Reporter signature ions are released during the MS2 fragmentation of tagged peptides, yielding the simultaneous identification and relative quantification of the peptides among multiple samples. The key feature of the TMT labeling approach is that it can afford a high degree of multiplexing, since it is able to analyze up to 16 samples in a single analysis. The intensities of each mass tag reflect the relative concentration of the original labeled peptide in each of the samples. TMT multiplex, which contains a maximum of 16-plex currently, has been widely implemented in comparative proteomics for its superior features of high quantitative accuracy and precision, and has been used for global discovery in serum proteomics [98]. TMTs have been heavily applied for various samples such as cells, tissues, and body fluids in shotgun proteomics for cancer research [99,100].

Selected reaction monitoring (SRM), a most common method for targeted quantitative proteomics, is copied from an original small molecule analysis. SRM is often used for the absolute quantitation of peptides/proteins, as it is highly sensitive, reproducible, and robust, offering a high accuracy and precision (Figure 1) [101,102]. Generally, the SRM-based quantitation required a triple quadrupole (QQQ) mass spectrometer where the precursor molecular ions are selected in Q1 and fragmented in Q2 collisional cells, and a unique fragment ion is monitored in Q3. The absolute quantitation of a targeted peptide selected from a previously identified target protein is obtained by the use of a stable isotope labeled synthetic peptide serving as an internal standard (IS). The targeted peptide and the synthetic peptide are co-eluted in LC-MS/MS, and the ratio of the chromatographic peak area for the target/IS peptides is determined and used for absolute quantitation. A calibration curve of the SRM response of each synthetic native peptide versus its series of concentrations was generated. This calibration curve is central to the design, validation, and application of SRM assays. The high sensitivity and wide dynamic range, in combination with the inherent selectivity of MS, make SRM an ideal tool in the targeted quantitative analysis of complex clinical samples for cancer diagnosis [103,104].

Post-translational modifications (PTMs), including phosphorylation, acetylation, ubiquitination, and methylation, are identified and validated as critical for signaling transduction, protein degradation, and transcriptional regulation. PTMs are the main players for protein functions, and they contribute to abnormal cellular proliferation, adhesion characteristics, and morphology in cancer. However, there are limited published studies targeting PTMs in serum/plasma for cancer biomarkers [105]. The high abundance of proteins in the serum/plasma and generally low occupancy rate for most PTMs make it particularly challenging for the detection of low-abundance proteins, and most likely even harder for their PTMs. Therefore, a particular PTM enrichment workflow is needed [106]. The enrichment method allows PTM profiling from a reasonable volume of serum/plasma (~250 μL for multiple PTM enrichment), followed by LC-MS/MS analysis. The ability to acquire high-content quantitative information, determine the state of modification, and study protein–protein interactions for thousands of proteins in native biological samples provides key insights into the composition, regulation, and function of protein complexes and pathways, which allows us to develop a better understanding of cellular and physiological processes and disease mechanisms. Recently, new approaches to the proteomic profiling of cell type-specific proteins in circulating exosomes in serum have been developed to provide new diagnostic tools and monitor therapeutic response in cancer-related research [107,108]. Data-independent acquisition (DIA) is a relatively new MS data acquisition technology emerging in proteomics where each of the precursor ions are selected sequentially for co-fragmentation with a predefined m/z range [109,110,111]. Its untargeted and systematic sampling process enables the acquisition of comprehensive LC−MS data, consisting of all precursors and product ions present in the sample within the m/z range measured. DIA adds some unique advantages compared to traditional data-dependent acquisition (DDA) in proteomics, where only one selected precursor ion at a time over particular threshold is used for fragmentation. The main challenge for the conventional DDA-based quantification is the dynamic range of quantification, as a single survey of entire co-eluting ions is used for the quantification. In addition, DDA often fails to detect some of the relatively medium- or low-abundance peptides/proteins due to its limited acquisition cycle time for complex samples. In the case of DIA, theoretically all precursor ions are co-fragmented and have never been missed. More importantly, the DIA data can be retrospectively analyzed in a pseudo targeted fashion by extracting the co-eluting fragment ions corresponding to their precursor peptides of interest for both quantitation and the verification of identity [112,113]. DIA-based quantitation adds the MS2 measurement, which is expected to be better than MS1 alone in particularly complex matrices due to its better selectivity and lower background noise [114]. The main downside of DIA is that, in most cases for global proteome application, an upfront generation of a spectra library is often required by DDA analysis on the fractionated complex samples for the confident identity of peptides/proteins due to the complexity of MS2 co-fragmentation in DIA data. Recently, DIA workflows are used for proteomic quantitative analysis for biomarker discovery in cancers using plasma, tissue, or other body fluids [115,116].

The main limitations and bottlenecks to be overcome in this field are reducing the inter and intra- individual variability of the samples without impacting the depth of the proteome coverage. This requires studies with adequately large numbers of samples from multiple independent cohorts to be analyzed to achieve validation. Large improvements in sample preparation methods for blood-based proteomics are still needed for those very low-abundance biomarkers, particularly for serum samples with an extremely wide dynamic range of proteins. Another challenge is to effectively integrate the proteomic data into clinical applications. For example, the need to determine the exact role of candidate proteins discovered in quantitative proteomics as validated and useful diagnostic, prognostic, predictive, or resistance biomarkers with rapid turnaround times for patient care for any specific clinical application remains a challenge.

## 6. Clinical Serum/Plasma Proteomics and Molecular Signatures of Cancers

The unravelling of cancer mutational landscapes has enabled the development of targeted therapies called New Molecular Entities (NMEs). In the past decade, multiple drug choices for the same tumor type in the same tumor stage have emerged and created an urgent need for identifying predictive molecular signatures of response and/or resistance, since none of the NMEs are effective in all patients and are linked to causing drug toxicities in some patients. Ongoing research efforts have recently focused on the integration of “multi-omics”, which include genomic and genetic alterations along with serum proteomic biomarkers into a single platform or signature for clinical application to match drugs to a patient’s individual molecular profile for increasing the effectiveness and limiting harm.

An example of an integrated multi-omics algorithm-based approach for cancer screening that uses mutational alterations and proteins is the CancerSEEK clinical test. This test focuses on driver mutations in key genes detected in cell-free DNA from blood, along with key proteins detected in blood as a screening test for the earlier detection of cancers of the ovary, liver, stomach, pancreas, esophagus, colorectal, lung, or breast. A similar integration of serum proteomic biomarkers for screening in patients with an increased cancer risk based on their family history or inherited disorders such as Lynch Syndrome characterized by mutations in genes such as the DNA mismatch repair genes (MLH1, MSH2, MSH6, PMS1, PMS2) [117], TP53-associated cancers in the case of Li–Fraumeni syndrome, and BRCA1/2 for breast cancer is currently lacking. Similarly, post-cancer diagnosis genomic and proteomic companion diagnostics are currently being explored for staging, prognostication, and choosing specific therapies for practicing precision medicine [118], but have not yet changed clinical practice. Clinical research and the applications of polygenic risk scores [119] continue to be refined for molecular prognostication [120] for predicting response to treatments (molecular predictive biomarkers), predicting the recurrence of cancer after initial treatments [121], predicting the lack of response to specific therapy (markers of primary resistance), and monitoring minimal residual disease (MRD) after treatments. A comprehensive list of tumor types in the last 10 years of publications with clinical applications based on a wide variety of proteomic techniques on blood-based fluid is summarized in Table 2. In this section, an overview of the top four tumor types from the list is highlighted.

### 6.1. Lung Cancer

Lung cancer has the highest cancer-related mortality in both men and women [163]. The overall 5-year survival rate for lung cancer is 19%, and late diagnosis is a major obstacle in improving lung cancer prognosis [164,165]. The five-year survival rate for the early lung cancer patients is 80%, suggesting that early detection is an important management tool for improving outcomes. Screening for lung cancer using cancer biomarkers therefore has immense value. Some of the serum/plasma tumor markers recently developed for the detection, diagnosis, prognosis, or management of lung cancers include cytokeratin 19 fragments (CYFRA 21-1), carcinoembryonic antigen (CEA), epidermal growth factor receptor (EGFR), soluble mesothelin-related proteins (SMRP), and Osteopontin (pOPN) [123,124,166,167]. Table 2 summarizes the ongoing research efforts on potential markers that use a variety of proteomic technologies. Additionally, both serum- and plasma-based profiling have been investigated for the screening and diagnosis of lung cancers [168]. Nevertheless, the validation of promising candidate markers in a large set of samples remains to be performed.

### 6.2. Breast Cancer

Breast cancer (BCa) is the most common cancer in women, resulting in about 15% of deaths in females [163]. Several advancements in breast cancer screening and diagnosis, novel therapeutic drugs, and combination treatments with radiation and surgical techniques have resulted in an increase in the 5-year survival rate to up to 90% if the cancer is detected early and treated appropriately [169]. Mammography is one of the most important screening tools as a population strategy, but has limitations in young women (under 40 years old) or women with large breasts, with a sensitivity of as low as 25–59% and high rates of false negative and false positive in diagnosis [170]. Blood-based proteomics as a screening diagnostic with novel noninvasive biomarkers for the early detection of BCa in asymptomatic individuals for the diagnosis and prognosis of the disease states has been an area of research interest [171,172]. Promising candidate biomarkers such as CA125, carcinoembryonic antigen (CEA), oncogenic protein RS/DJ-1, human epidermal growth factor receptor-2 (HER2), and circulating cytokeratin fragments (tissue polypeptide antigen (TPA), tissue polypeptide-specific antigen (TPS), and CYFRA 21-1) have been identified, but are pending validation [173]. Several of these are less specific for breast cancer or have a low sensitivity [173,174,175]. For example, a set of 10 potential BCa serum biomarkers and cancer antigens (haptoglobin, osteopontin (OPN), CA15-3, CA125, carbohydrate antigen 19-9 (CA19-9), CEA, prolactin, α-fetoprotein (AFP), leptin, and migration inhibitory factor (MIF)) were developed for diagnosis and screening, but none of them are detected to have a high specificity, particularly in detecting early stage disease [176,177]. Other plasma candidate biomarkers available used for the screening and diagnosis of BCa include serpin peptidase inhibitor (SERPINB4), secreted-clusterin (CLU), serum amyloid (SAA), and heat shock proteins (HSPs) [128,129,178,179], but have yet to be validated in large studies for population application strategies. The majority of the markers reported by single breast cancer studies are based on a limited number of samples, hence the validation of biomarker candidates by the targeted profiling analysis of large cohorts of samples and populations is crucial for implementing those in clinical application.

### 6.3. Ovarian Cancer

Ovarian cancer is one of the top five causes of cancer deaths in women, with over 21,000 new diagnoses and nearly 14,000 deaths expected in 2020 in the United States alone [163]. The 5-year survival rate for those diagnosed with distant disease is approximately 30% compared to nearly 93% for those diagnosed with localized disease [163,180]. Thus, early detection and diagnosis has potential for a profound impact on the survival rate [181,182]. For general population screening, early detection markers with a high sensitivity and extremely high specificity are needed to maximize the number of cases detected, while at the same time avoiding great numbers of false positive results [32,183,184]. Cancer Antigen 125 (CA125), a glycoprotein, is by far the most studied and useful single biomarker until today for the diagnosis of serous and endometroid epithelial ovarian carcinomas, but it is non-specific due to its presence in other benign gynecological and abdominal conditions [185,186,187,188]. The sensitivity of CA 125 in distinguishing between benign and malignant masses ranges between 61% and 90%, while the specificity ranges between 35% and 91% [189]. Thus, in spite of its strengths, improvements can be made. It is unlikely that a single marker can achieve the needed discrimination due to the complexity and heterogeneous nature of ovarian cancer. A combination of biomarkers may provide a higher specificity and sensitivity for the early detection of ovarian cancer [190]. Serum proteomics profiling is widely used for discovery of such markers, offers a high level of information across a large group of heterogeneous patients, and has been proven to be more likely to contain successful markers than the single low molecular weight serum biomarker [191,192]. Several potential serum/plasma biomarkers for the early detection and disease monitoring of ovarian cancer have been reported [80,135,193,194]. In ovarian cancer, the identification of biomarkers in serum/plasma is difficult even if they are highly expressed in plasma, since ovarian cancer is characterized by loss-of-function mutations and the downregulation of tumor suppressor activities [136]. Nonetheless, detecting those low-abundance proteins by mass spectrometry is extremely challenging from a technical standpoint, with high levels of glycosylation. Recently, the so-called mass-spectrometry-based quantitative proteomics has become the common strategy for identifying proteins and their alterations [195]. Hence, our hope is that additional studies will be performed adhering to the study design principles described herein to discover novel serum protein signatures to improve not only the early detection of ovarian cancer, but also the guidance on ovarian cancer therapy. Biomarker research is now focusing on the discovery of different biomarkers either in serum or plasma expressed in different stages and different subtypes of the disease [17,196,197,198].

### 6.4. Prostate Cancer

Prostate cancer is one of the most common malignancies among men, with a global incidence of 1.3 million new cases every year [199]. While prostate cancer may require minimal or even no treatment at early stages, metastasizing to the bone is a serious concern for about 65–80% of advanced stage cases [200,201]. The five-year survival rate of prostate cancer patients with localized prostate cancer has increased to over 90% with improved technologies for the treatment at an early stage; however, once the tumor cells metastasize, the 5-year survival rate substantially decreases to only 28% [202]. Prostate-specific antigen (PSA) is a serine protease that has been used extensively in clinical practice since 1988. The discovery of PSA as a serum biomarker for prostate cancer revolutionized its usage for monitoring disease recurrence after initial treatment and for evaluating response to cancer treatments [203]. However, the serum levels are elevated from both malignant and non-malignant causes and furthermore, cannot discriminate lethal from indolent prostate cancer. A significant up-regulation of periostin, a serum biomarker, was found in prostate cancer patients for prognosis, but is also a potential target for therapeutic intervention [204]. Larkin et al. found with serum profiling via iTRAQ labeling that the combination of Pre-rRNA-processing protein TSR1 along with PSA improved the diagnostic performance [139]. In advanced stage disease, where androgen deprivation therapy (ADT) remains a cornerstone intervention, predictive biomarkers are not well established. Recently, using an iTRAQ-based serum proteomic profiling in 50 serum specimens from 30 advanced prostate cancer patients, Kohli et al. discovered some response biomarkers to ADT. An initial discovery set of 47 potential serum biomarkers associated with ADT response factors during the treatment included small molecules such as elevated beta estradiol and estrone sulphate levels [22]. Several potential biomarkers recently found in either serum/plasma need further validation [138,140,205]. However, the validation of this small study including longitudinal follow-ups of clinical outcomes is needed. Other proteomic approaches, including the use of cell lines or animal models for discovery, followed by validation in clinical specimens, have demonstrated early promise for discriminating between cancer and healthy patients, or as markers of [206,207,208], but validation cohorts will be needed for confirming the value of serum/plasma proteomic-based biomarkers in clinical settings.

## 7. Conclusions

Proteomics approaches may enhance biomarker research in cancer, but challenges remain. A step-wise approach to meeting pre-analytic and analytic challenges and an a priori statistical and data analysis approach can yield a huge amount of information. Proteomics provides an opportunity for a relatively non-invasive and sensitive tool for identifying characteristic proteins and peptides, which may allow the potential for serial disease monitoring. Meeting these biological and technical challenges is critical for synthesizing analytical study designs and future research. These challenges can be overcome by using statistical methodologies incorporated in proper study designs, implementing strict protocols for specimen biobanks and sample handling, establishing robust validation assays, and exploring innovative tools. Prior to and/or during the discovery design phase, the sensitivity, the specificity of detection, the downstream clinical application for the development of the biomarker, and the population of interest are critical determinants for successful proteome biomarker development. Technical innovations such as increasing the reproducibility, minimizing the variability, improving strategies for quality control, increasing the sample throughput. and detecting various types of protein alterations would help biomarker discovery efforts. New developments in mass spectrometry instruments and recently developed quantitative reagents such as TMT 16-plex help to include a larger number of samples, resulting in high throughput and increased data reliability, and are expected to accelerate proteomic biomarker discovery. As a result, following validation analysis against large cohort samples, a growing number of blood-based biomarkers are expected to become cancer signatures used for clinical application in the diagnosis and prognosis of disease states and predictive response to specific therapies. In the future, continuous advances in targeted MS technologies and applications for enhanced sensitivity, accuracy, and throughput will revolutionize the field of clinical research by measuring patients’ protein concentrations in large numbers of clinical specimens. Enhancing the sensitivity for in-depth proteome coverage can be also achieved by either simplifying or increasing the efficiency of the front-end sample preparation steps.

## Figures and Tables

**Figure 1 cancers-12-02428-f001:**
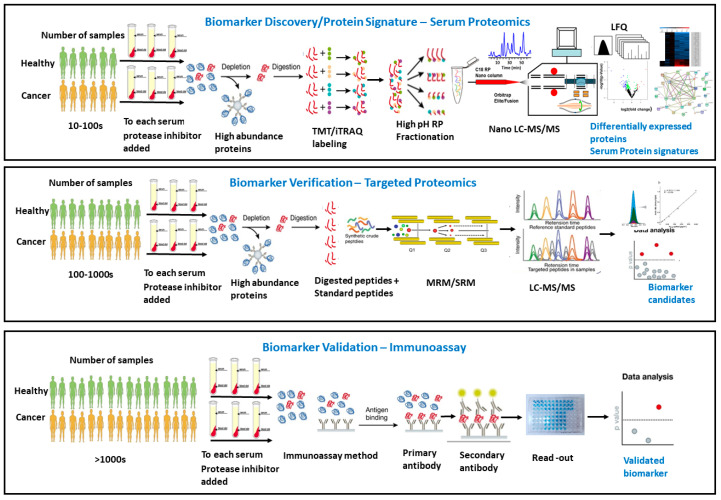
Current serum proteomics workflows for the discovery, verification, and validation of cancer biomarkers. Top panel: global comparative serum proteomics workflow for screening some candidates of protein signature from tens/hundreds of differentially expressed proteins found between healthy and cancer patients. Middle panel: targeted quantitative serum proteomic workflow in a larger number of samples for verifying potential biomarker candidates. Bottom panel: the development of an immunoassay for selected potential biomarkers for validation and clinical application.

**Table 1 cancers-12-02428-t001:** Experimental design and statistical considerations for maximizing the likelihood of reproducible biomarker discovery findings.

Study Design Step	Statistical Considerations
**Study Goal**	Define the population in which the protein marker will be used. Define the purpose of the protein marker—e.g., early detection, disease monitoring, etc.
**Specimens**	Select case and control specimens randomly. Select from a prospectively collected (prior to knowledge of disease status) specimen biobank. Select specimens from the relevant time point in the disease course. Avoid convenience samples. Avoid pooling of specimens.
**Study Design**	Plan sufficient sample size for discovery in light of realistic expected differences. Randomize specimens to assay run order.
**Differential Abundance Detection**	Assess protein difference signals relative to variation. Incorporate statistical design into the analysis model. Apply correction for multiple comparisons.
**Panel or Signature Model Building**	Finalize analysis plan in writing prior to beginning analyses. Employ optimism correction methods. Generate a fixed, locked-down algorithm.
**Validation**	Perform verification of initial protein marker identifications. Perform internal model validation in the discovery sample set. Perform external model validation in an independent sample set.

**Table 2 cancers-12-02428-t002:** Clinical applications of blood-based proteomic profiling in cancer.

Tumor Type	(Serum/Plasma)-Proteomics Method	Biomarker Identified	Clinical Application in Cancer	Stage of Tumor	References
**Lung**	Plasma-MALDI-TOF, PRM	EGFR, SPRAC, SAA1, SAA2	Screening/Prognostic	Advanced/Local	[122,123,124]
	Serum-Shotgun	PTM profiling, (Hp) β chain	Screening	Advanced	[125]
	Plasma-Shotgun	Cytokines, Peroxiredoxins	Diagnosis/screening	local	[126]
**Breast**	Serum–QTOF LC-MS	PRG4, C1-inh	Screening	local	[127]
	Plasma-2Dgel, shotgun	CLU, SAA, SERPINB4, COL11A1	Screening	Advanced	[128]
	Plasma-iTRAQ	THBS1, BRWD3, EGFR, CFHR3	Prognostic	local	[129]
**Colorectal**	Plasma-MRM-MS	MASP1, OPN, PON3, TFRC	Screening	local	[130,131]
	Plasma-MRM-MS	270 protein biomarkers	Diagnosis	local	[132]
	Serum-LC-MS/MS, 2Dgel	Reporter peptides, NDKA	Screening/prognostic	Advanced	[133,134]
**Ovarian**	Plasma-SRM	CA125, IGHG2, LGALS3BP, DSG2, L1CAM, THBS1	Diagnosis	Advanced	[53]
	Serum-MALDI-TOF, LC-MS/MS	PTM profiling, RBP4	Diagnosis	Local/Advanced	[49,135]
	Serum-iTRAQ, MRM	Protein Z, CA125, CLIC4, TPM	Screening, Diagnosis	Local	[136,137]
**Prostate**	Serum-, iTRAQ	sE-cadherin, TSR1, SAA, KLK3	Diagnosis	Local	[138,139]
	Plasma-, Shotgun	CA1	Diagnosis	Advanced	[140]
**Colon**	Serum-iTRAQ, MRM	LCN-2, CELA1, CEL2A, CTRL, LRG1, TUBB5	Prognostic/Screening	Local/Advanced	[126,141,142,143]
	Plasma-, MRM	SLeA, TIMP1, COMP, THBS2, MMP9	Prognostic/Screening	Local/Advanced	[126,131,141,142]
	Serum-Shotgun	IGJ, APOA1, PCOLCE, SAA2	Prognostic	Local	[131,144,145]
**Lymphoma**	Serum-Shotgun	SOX3, CA9, MSLN, CCL20, SCF, MMP-10, IGF-1, TRIM3	Prognostic/Diagnosis	Local/advanced	[144,145,146,147,148]
**Gastric**	Serum-iTRAQ, MALDI-TOF	GRN, 14-3-3β	Prognostic/Diagnosis	Local	[146,147,148,149,150]
	Serum-SEC-Shotgun, MALDI-TOF	YWHAG, RBM6, LAMC2	Prognostic/Diagnosis	Local/Advanced	[149,150,151,152,153]
**Pancreatic**	Serum-MRM	apoA-IV, apo-CIII, IGFBP-2 ICAM-1, TIMP-1	Diagnosis	Advanced	[151,152,153,154,155]
	Plasma-LC-MS/MS	F9, CFI, AFM, HPR, ORM2	Prognostic	Local	[154,155,156]
**Cervical**	Serum-2Dgel	MMP	Diagnosis	Local/Advanced	[156,157]
	Serum-MALDI-TOF	Peptide profiling	Diagnosis	Local	[157,158]
	Plasma-iTRAQ, LFQ	TXN, TXNDC5	Screening	Local	[158,159]
**Multiple Myeloma**	Serum-LC-MS/MS	SAA2, KLKLB1, APOA1, CD44	Diagnosis	Advanced	[159,160,161]
	Serum-MALDI-TOF	Peptide Profiling	Diagnosis	Local	[160,161,162]

MALDI-TOF: Matrix-assisted laser desorption ionization-time of flight; QTOF: Quadrupole time-of-flight; PRM: Parallel reaction monitoring; 2D-gel: Two dimensional gel electrophoresis; iTRAQ: Isobaric tag for relative and absolute quantitation; LC-MS/MS: Liquid chromatography tandem mass spectrometer; MRM: Multiple reaction monitoring; SEC: Size exclusion chromatography; LFQ: Label-free quantitation; PTM: Posttranslational modifications.

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
