# Peer review of "Challenges and Opportunities in Clinical Applications of Blood-Based Proteomics in Cancer"

_cancers, 2020, doi:10.3390/cancers12092428_

Round 1

Reviewer 1 Report

The review from Bhawal et al. introduced the entire workflow from sample collection to quantitative proteomics and data analysis as well as summarized the literatures that applied blood proteomics analysis for cancer biomarker discovery. The field is important; the recent advancements in proteomics technology make it a promise role in both identification and verification (or even validation) of potential disease markers. However, few of these more recent technologies were included and many inadequate citations and wording errors were found in this review. Following are several suggestions:

  1. Proteomics study design: in order to have a precise quantitation result that can unbiasedly reveal the molecular differences between two or multiple cohorts and filter the differentially expressed proteins with high sensitivity, it is very important to have individualized serum/plasma proteome analysis. But the authors did not emphasize this issue in most of the described literatures and suggest to use the pooled serum (as shown in Figure 1, upper and middle panels) for biomarker discovery and verification. Please revise the content.
  2. Label free quantitation: Please add illustration for label free quantitation in discovery phase in Figure 1 (upper panel). In page 8, line 319-325, add the construction of calibration curve by using synthetic light and heavy (isotope) peptides for absolute quantitation (can add this to Figure 1, verification panel too).
  3. New MS technology: the authors describe mainly the data-dependent acquisition workflow as MS detection of peptide or proteins. However, several papers already demonstrated the use of data-independent acquisition method for more comprehensive profiling of blood proteome in individual serum/plasma samples. Please add this new advancement and related clinical applications in the manuscript.
  4. Inadequate and incorrect citations: Please examine whether you have correct citations for all the papers. In addition, please cite the original papers, especially when you describe the observed biological effects or technology. For example, ref [1] is not for genetic and epigenetic alterations to promote tumorigenesis, [65] is a paper from genome analysis, not for proteome analysis, [93,94] should be the original papers for iTRAQ and TMT. [106] described the detection of EBV DNA in blood for nasopharyngeal cancer, not the use of blood protein analysis for multiple cancer types.
  5. Table 2: please thoroughly examine the documentation and citation of papers in this table. Please remove the review citations and cite the original works to acknowledge the contribution of other authors. Taking the lung tumor type as example, refs. [115, 116, 120, 176] are review papers. In addition, ref [175, 179] are papers using ELISA, not proteomics method. Remove the serum/plasma-ELISA category from this table, because it is not a proteomic method. For other cancer types, similar errors were found.
  6. Add one paragraph to describe the limitations or bottlenecks remain to be overcome in this field.

Minor errors:

  1. Please correct the “et al.” throughout the manuscript
  2. Line 100, revise to “General considerations for serum/plasma proteomics-based biomarker discovery”
  3. Line 107, add “.” After [25].
  4. Line 175, delete space between #°C

Author Response

Reviewer 1:
Comments and Suggestions
“The review from Bhawal et al. introduced the entire workflow from sample collection to quantitative proteomics and data analysis as well as summarized the literatures that applied blood proteomics analysis for cancer biomarker discovery. The field is important; the recent advancements in proteomics technology make it a promise role in both identification and verification (or even validation) of potential disease markers. However, few of these more recent technologies were included and many inadequate citations and wording errors were found in this review. Following are several suggestions:
Proteomics study design: in order to have a precise quantitation result that can unbiasedly reveal the molecular differences between two or multiple cohorts and filter the differentially expressed proteins with high sensitivity, it is very important to have individualized serum/plasma proteome analysis. But the authors did not emphasize this issue in most of the described literatures and suggest to use the pooled serum (as shown in Figure 1, upper and middle panels) for biomarker discovery and verification. Please revise the content.”
Reply to Comment 1:
We thank the reviewer for the appreciation on the review. We appreciate the reviewer’s suggestions/comments on proteomic study design. As a result, we now mention the limitation of pooled serum for biomarker discovery and verification between lines 213-215 in tracked changes.
“Pooling of specimens prior to mass analysis without the specimens being labeled should be avoided as it relies on the assumption of biological averaging and it constrains the types of analyses that can be performed.” Line 213-215 is the new input and line 210 we add “or to pool specimens”.
We have also added the entry “Avoid pooling of specimens” to Table 1 as the last item under “Specimens”, and have also replaced the previous Figure 1 with a new Figure that reflects this change.
Specifically, in Figure 1 for each one of the panel “serum pooled” is now replaced by “serum”.

Comment 2:
“Label free quantitation: Please add illustration for label free quantitation in discovery phase in Figure 1 (upper panel). In page 8, line 319-325, add the construction of calibration curve by using synthetic light and heavy (isotope) peptides for absolute quantitation (can add this to Figure 1, verification panel too).”
Response to Comment 2:
Thank you for this suggestion.
In page 8, line 326-329, we have now added “A calibration curve of the SRM response of each synthetic native peptide versus its series of concentrations was generated. This calibration curve is central to the design, validation, and application of SRM assays”.
Additionally, in Figure 1, “the calibration curve diagram was added in the verification panel”. The replaced Figure 1 has this change accommodated as well.

Comment 3:
“New MS technology: the authors describe mainly the data-dependent acquisition workflow as MS detection of peptide or proteins. However, several papers already demonstrated the use of data-independent acquisition method for more comprehensive profiling of blood proteome in individual serum/plasma samples. Please add this new advancement and related clinical applications in the manuscript.”
Response to Comment 3:
Thank you for pointing this out. We agree with the suggestion and have added this to the text on Page 9, lines 348-369 reflects the changes made as follows:
“Data independent acquisition (DIA) is a relatively new MS data acquisition technology emerging in proteomics where each of the precursor ions are selected sequentially for co-fragmentation with a predefined m/z range [108-110]. Its untargeted and systematic sampling process enables acquisition of comprehensive LC−MS data consisting of all precursors and product ions present in the sample within the m/z range measured. DIA adds some unique advantages compared to traditional data dependent acquisition (DDA) in proteomics, where only one selected precursor ion at a time over particular threshold was used for fragmentation. The main challenge for the conventional DDA-based quantification is the dynamic range of quantification, as a single survey of entire co-eluting ions is being used for the quantification. In addition, DDA often fails to detect some of the relatively medium or low abundance peptides/proteins due to its limited acquisition cycle time for complex samples. In case of DIA, theoretically all precursor ions are co-fragmented and have never been missed. More importantly, DIA data can be retrospectively analyzed in a pseudo targeted fashion by extracting the co-eluting fragment ions corresponding to their precursor peptides of interest for both quantitation and verification of identity [111,112]. DIA-based quantitation adds the MS2 measurement, which is expected to be better than MS1 alone in particularly complex matrices due to better selectivity and lower background noise [113]. The main downside of DIA is that in most cases for global proteome application, an upfront generation of spectra library is often required by DDA analysis on the fractionated complex samples for confident identity of peptides/proteins due to the complexity of MS2 co-fragmentation in DIA data. Recently, DIA workflows are used for proteomic quantitative analysis for biomarker discovery in cancers using plasma, tissue or other body fluids [114,115]”.

Comment 4:
“Inadequate and incorrect citations: Please examine whether you have correct citations for all the papers. In addition, please cite the original papers, especially when you describe the observed biological effects or technology. For example, ref [1] is not for genetic and epigenetic alterations to promote tumorigenesis, [65] is a paper from genome analysis, not for proteome analysis, [93,94] should be the original papers for iTRAQ and TMT. [106] described the detection of EBV DNA in blood for nasopharyngeal cancer, not the use of blood protein analysis for multiple cancer types.”
Response to Comment 4:
We thank the reviewer for this correction. As a result the following changes to the references have been made in this resubmission:
ref[1] was removed, [65] was removed, [93,94] was in the references for TMT and iTRAQ, [106] reference was removed, most of the review references were replaced by the original papers.

Comment 5:
“Table 2: please thoroughly examine the documentation and citation of papers in this table. Please remove the review citations and cite the original works to acknowledge the contribution of other authors. Taking the lung tumor type as example, refs. [115, 116, 120, 176] are review papers. In addition, ref [175, 179] are papers using ELISA, not proteomics method. Remove the serum/plasma-ELISA category from this table, because it is not a proteomic method. For other cancer types, similar errors were found.”
Response to Comment 5:
We went through Table 2 as suggested and have removed all the review citations to cite the original publications only. We have also removed the serum/plasma- ELISA category from Table 2 for all cancer types.

Comment 6:
“Add one paragraph to describe the limitations or bottlenecks remain to be overcome in this field.”
Response to Comment 6:
A new paragraph to describe the limitation or bottlenecks remain to be overcome in this filed were added in page 9, line 370-379.
“The main limitations and bottlenecks to be overcome in this field are to reduce inter and intra- individual variability of the samples without impacting the depth of proteome coverage. This requires studies with adequately large number of samples from multiple independent cohorts to be analyzed to achieve validation. Large improvements in sample preparation methods for blood-based proteomics are still needed for those very low abundance biomarkers particularly for the serum samples with the extremely wide dynamic range of proteins. Another challenge is to effectively integrate the proteomic data into clinical applications. For example, the need to determine the exact role of candidate proteins discovered in quantitative proteomics as validated and useful diagnostic, prognostic, predictive or resistance biomarkers with rapid turnaround times for patient care for any specific clinical application remains a challenge.”
Minor comments:
Minor errors:
1. Please correct the “et al.” throughout the manuscript
Response to Minor Comment
We have corrected this throughout the manuscript.
2. Line 100, revise to “General considerations for serum/plasma proteomics-based biomarker discovery”
Response to Minor Comment
We have changed the title in line 100 as suggested.
3. Line 107, add “.” After [25].
Response to Minor Comment
We have changed the line 107, by adding “.” after [25].
4. Line 175, delete space between #°C
Response to Minor Comment
We have deleted the space between #°C.

Reviewer 2 Report

I found this review article is well written. Though, author`s needs to thoroughly revise the manuscript for any typo-errors before it consideration for publication.

Author Response

Reviewer 2:
Comments and Suggestions for Authors
“I found this review article is well written. Though, author`s needs to thoroughly revise the manuscript for any typo-errors before it consideration for publication.”
Response to Comment
We thank the reviewer’s comment. We have gone through the manuscript to remove typographical errors in this resubmission.

Reviewer 3 Report

The review entitled “Challenges, Opportunities in Clinical Applications of Blood-Based Proteomics in Cancer” written by Ruchika Bhawal and coworkers, want to highlight the importance of recent advances in MS-based proteomics technologies to identify novel potential biomarkers for clinical applications. In my opinion the paper is well written and could be accepted after minor revision.

Only few recommendation:

The 2.1 paragraph “Choice of blood plasma or serum for proteomics studies” needs revision, is too long in this context and is not informative to focalize the attention on biomarkers; probably could be linked and joined with the following paragraph (2.2)

Line 107 miss dot

Table 1 please improve the formatting and make the table more readable. It is too full and unclear.

Author Response

Reviewer 3:
Comments and Suggestions for Authors
“The review entitled “Challenges, Opportunities in Clinical Applications of Blood-Based Proteomics in Cancer” written by Ruchika Bhawal and coworkers, want to highlight the importance of recent advances in MS-based proteomics technologies to identify novel potential biomarkers for clinical applications. In my opinion the paper is well written and could be accepted after minor revision.”
Comment 1
“The 2.1 paragraph “Choice of blood plasma or serum for proteomics studies” needs revision, is too long in this context and is not informative to focalize the attention on biomarkers; probably could be linked and joined with the following paragraph (2.2)”.
Line 107 miss dot
Response to Comment 1
We have changed the line 107, by adding “.” after [25]. As per the initial submission, the Editor on the Special Issue on ‘cancer Proteomics”, a specific request was made by Dr. Sam Hanash to us for including a section on serum versus plasma based proteomics. We agreed with the Dr. Hanash’s suggestion and expanded the rationale for serum versus plasma based cancer proteomics in Section 2.1.

Comment 2:
Table 1 please improve the formatting and make the table more readable. It is too full and unclear.
Response to Comment 2:
We have improved the formatting by shortening each point in the table to make it more readable and clear.